# Intronic ATTTC repeat expansions in *STARD7* in familial adult myoclonic epilepsy linked to chromosome 2

Mark A. Corbett ⓘ et al.[#]

Familial Adult Myoclonic Epilepsy (FAME) is characterised by cortical myoclonic tremor usually from the second decade of life and overt myoclonic or generalised tonic-clonic seizures. Four independent loci have been implicated in FAME on chromosomes (chr) 2, 3, 5 and 8. Using whole genome sequencing and repeat primed PCR, we provide evidence that chr2-linked FAME (FAME2) is caused by an expansion of an ATTTC pentamer within the first intron of *STARD7*. The ATTTC expansions segregate in 158/158 individuals typically affected by FAME from 22 pedigrees including 16 previously reported families recruited worldwide. RNA sequencing from patient derived fibroblasts shows no accumulation of the AUUUU or AUUUC repeat sequences and *STARD7* gene expression is not affected. These data, in combination with other genes bearing similar mutations that have been implicated in FAME, suggest ATTTC expansions may cause this disorder, irrespective of the genomic locus involved.

*email: jozef.gecz@adelaide.edu.au. A full list of authors and their affiliations appears at the end of the paper.

FAME (also referred to as Familial Cortical Myoclonic Tremor and Epilepsy or Benign Adult onset Familial Myoclonic Epilepsy [OMIM phenotypic series: PS601068]) is characterised by cortical myoclonic tremor and overt myoclonic and later generalised tonic-clonic seizures (GTCS)[1]. Onset of symptoms occurs in the second to third decade with variable expressivity within and between families; anticipation has been noted in some families[1]. The frequency of GTCS varies from 15 to 100% in 22 different families reported here (Table 1)[2]. Seizures are typically controlled with anti-epileptic drugs for generalised epilepsies, although rarely individuals have drug resistant epilepsy. FAME has been mapped to four distinct chromosomal loci. Most families link to chromosomes 8q24[3] or 2p11.2-q11.2[4], with an additional two families mapping to chromosome 5p15.31-p15[5] and one to chromosome 3q26.32-q28[6]. There is one report of autosomal recessive FAME caused by mutation in *CNTN2* where the phenotype was disputed[7,8]. Candidate genes and variants that fall within these common linkage intervals have been suggested for chr2 (*ADRA2B*) and chr5 (*CTNND2*); however, none of these genes have been shown to be allelic in all FAME families with linkage to the same interval[1]. We previously showed using identity-by-descent mapping that there are at least four distinct founder loci linked to FAME2 (OMIM:607876) on chr2[9].

The genetic cause of FAME has long remained elusive. The cause of FAME1, which is linked to chr8 (OMIM:601068), has recently been shown to be a complex repeat expansion of pentameric TTTTA and inserted TTTCA repeats into the fourth intron of the *SAMD12* gene[10,11]. In the same study, *TNRC6A* (chr16) and *RAPGEF2* (chr4) were implicated as FAME genes within single families, respectively, found via direct detection of the same repeated TTTTA and TTTCA sequences[11].

Here, we use bioinformatic analysis of short-read whole-genome sequencing to identify ATTTT and ATTTC repeat expansions in the FAME2 linkage interval. We screen for an intronic ATTTC expansion in the first intron of STARD7 by repeat-primed PCR and show it segregates with FAME2 in 158 affected individuals from 22 families. We use long-read sequencing to suggest the ATTTT and ATTTC expansions may be somatically unstable. We analyse clinical data and show evidence of anticipation over multiple generations of a large FAME2 family. Finally, we demonstrate that the presence of the ATTTC repeat has no effect on protein or mRNA expression levels of STARD7 in available patient cell lines. These data suggest the repeat sequence alone is pathogenic, independent of an effect on the coding sequence of the encompassing gene.

## Results

**Discovery of a repeat expansion in STARD7.** We analysed Illumina HiSeq X-10 whole-genome sequencing data initially from two individuals from a large Australian-New Zealand FAME family, one from an Italian family and three from a French-Spanish family (Table 1 and Supplementary Table 1; Families 1, 3 and 19, respectively)[2,12,13] with two repeat expansion detection methods, ExpansionHunter and exSTRa[14,15], to look for similar combined ATTTT and ATTTC repeat expansions on both the forward and reverse chromosome strands within the FAME2 interval. This revealed an expansion of an ATTTT repeat and insertion of an ATTTC repeat in the context of the reverse strand of chr2 within the first intron of *STARD7* (StAR-related lipid transfer domain-containing 7) in all FAME samples tested (Fig. 1a, Supplementary Fig. 1). The endogenous ATTTT repeat in intron 1 of *STARD7* was also found to be variable in length in the normal population but not expanded to the same extent as repeats found in individuals with FAME. The ATTTC repeat was not present in any whole-genome sequencing data from 69 control

**Table 1 Clinical summaries of 22 investigated FAME families**

| Family | Nationality | Total affected | Mean onset [range] | Myoclonus/CT | TCS | Focal Sz | References |
|---|---|---|---|---|---|---|---|
| 1 | Australian/New Zealand of European ancestry | 55 | 18.6 y [4–59,60 y] | 55/55 (100%) | 8/55 (15%) | 2/55 (4%) | 2 |
| 2 | Italian | 2 | 15–25 y | 2/2 (100%) | 2/2 (100%) | 0/2 (0%) | |
| 3 | Italian | 4 | 2–18y | 4/4 (100%) | 4/4 (100%) | 2/4 (50%) | 12 |
| 4 | Italian | 11 | 22.3 y [12–49,50 y] | 11/11 (100%) | 11/11 (100%) | 3/11 (27%) | 4,17 |
| 5 | Italian | 25 | 26.6 y [5–39,40 y] | 25/25 (100%) | 10/25 (40%) | 0/25 (0%) | 38 |
| 6 | Italian | 12 (3 studied) | 12 y [8–17,18 y] | 11/12 (91.6%) | 6/12 (50%) | 1/12 (8.3%) | 9,39 |
| 7 | Italian | 4 | 22.75 y [10–35,36 y] | 4/4 (100%) | 3/4 (75%) | 0/4 (0%) | |
| 8 | Italian | 10 (6 studied) | 18.5 y [17–19,20 y] | 6/6 (100%) | 4/6 (66.6%) | 0/6 (0%) | 40 |
| 9 | Italian | 13 (11 studied) | 17 y [12–21,22 y] | 11/11 (100%) | 9/11 (81.1%) | 0/14 (0%) | 41 |
| 10 | Italian | 16 (14 studied) | 15.8 y [13–19,20 y] | 14/14 (100%) | 10/14 (71%) | 0/14 (0%) | 41 |
| 11 | Italian | 10 (5 studied) | 15.5 y [13–17,18 y] | 5/5 (100%) | 4/5 (80%) | 0/5 (0%) | 42 |
| 12 | Italian | 21 (17 studied) | 39.2 y [24–55,56 y] | 17/17 (100%) | 13/17 (76.4%) | 0/17(0%) | 43 |
| 13 | Italian | 3 | 17.7 y [12–22,23 y] | 3/3 (100%) | 1/3 (33%) | 0/3 (0%) | 42,44 |
| 14 | Italian | 3 | 16.3 y [15–18 y] | 3/3 (100%) | 3/3 (100%) | 0/3 (0%) | 44,45 |
| 15 | Italian | 4 | 30.3 y [18–48,49 y] | 4/4 (100%) | 3/4 (75%) | 2/4 (50%) | 17 |
| 16 | Iraqi of Sephardic Jewish ancestry | 15 (10 studied) | 21 y [12–31,32 y] | 10/10 (100%) | 4/10 (40%) | 2/10 (20%) | |
| 17 | Israeli of Sephardic Jewish ancestry | 2 | 21 y [21 y] | 2/2 (100%) | 2/2 (100%) | 0/2 (0%) | |
| 18 | South African of European ancestry | 24 (15 studied) | 15.8 y [11–19,20 y] | 15/15 (100%) | 7/15 (47%) | 1/15 (7%) | 46 |
| 19 | French/ Spanish | 13 | 41 y [30–59,60 y] | 13/13 (100%) | 8/13 (62%) | 0/13 (0%) | 13,47 |
| 20 | French | 7 (2 studied) | 20 y (n = 1) Childhood (n = 1) | 2/2 (100%) | 1/2 (50%) | 0/2 (0%) | 9 |
| 21 | Syrian | 1 | 20 y | 1/1 (100%) | 1/1 (100%) | 0/1 (0%) | |
| 22 | Italian | 11 (10 studied) | 25.1 y [14–39,40 y] | 9/10 (90%)a | 4/10 (40%) | 1/10 (10%) | |

*CT* cortical tremor, *Focal Sz* focal seizures, *TCS* tonic-clonic seizures, *y* years, *n* number of individuals
aOne family member last evaluated at 9 years of age

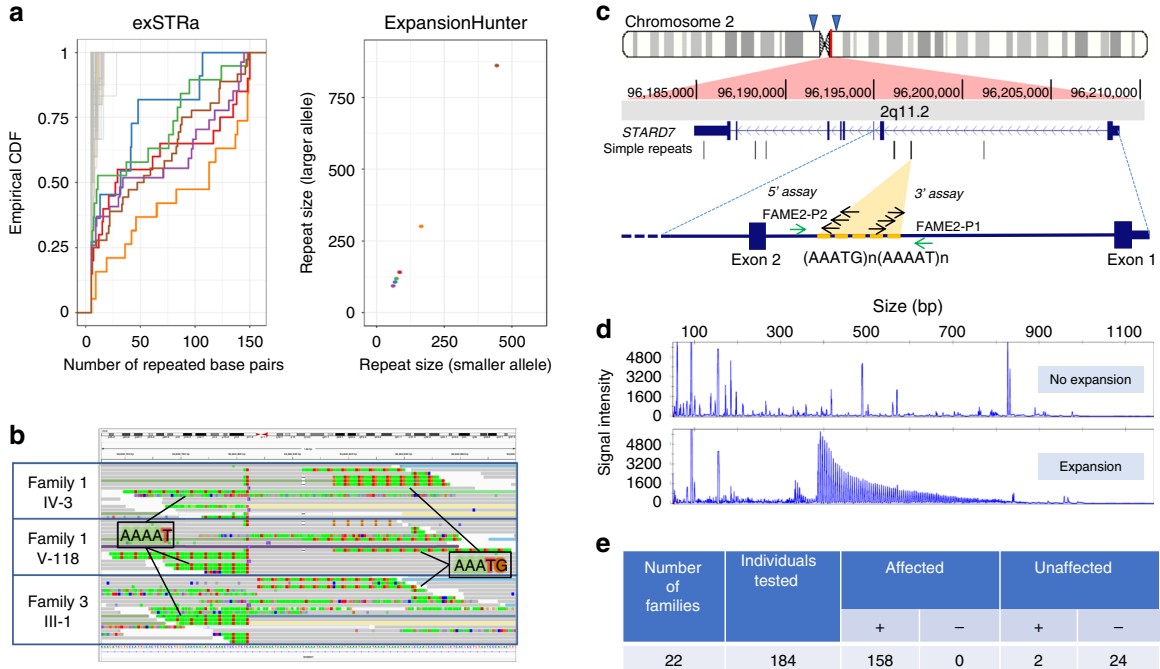

**Fig. 1** Identification of an expanded pentameric ATTTC repeat causing FAME2. **a** Estimated sizes of the AAATG repeats in two affected individuals from Family 1 (red, orange), one from Family 3 (brown) and three affected individuals from Family 19 (blue, green, purple), compared to 69 individuals without FAME using TruSeq Nano (grey) or KAPA Hyper (tan) library preparation. Left panel shows empirical cumulative distribution functions from exSTRa panel while the right panel shows the estimated repeat size by Expansion Hunter (the sum of both alleles suggests repeat sizes of 0.75–2.3 kb). Data underlying this part of the figure are available in Source Data. **b** WGS data from two individuals in Family 1 and one from Family 3 show reads suggesting expansion of AAAAT and insertion of AAATG repeats in the chr2 linkage interval. **c** Upper section shows the location of the repeat in the context of chr2. The approximate location of the FAME2 minimal linkage interval is shown above the ideogram with two blue arrow heads. The *STARD7* gene is on the reverse chromosome strand and the endogenous AAAAT repeat is found in the first intron of the gene. Schema in the lower section shows the primers used in the RP-PCR to detect the ATTTT "3′ assay" and ATTTC "5′ assay" expanded repeats, respectively. **d** Example results of the RP-PCR 5′ assay obtained in an individual negative for the ATTTC insert (top panel) and in an individual affected by FAME, positive for the ATTTC repeat insertion (bottom panel). Full screening results are provided in Supplementary Data 1. **e** Summary of 184 individuals from 22 families tested with the RP-PCR assay. Individuals under category (+) tested positive for the ATTTC repeat and individuals under category (−) tested negative for the repeat

samples (Supplementary Fig. 1), nor is it reported in the Simple Repeats track in the UCSC genome browser (build hg38)[16].

**Segregation of STARD7 ATTTC expansions by repeat-primed PCR.** We developed a repeat-primed PCR (RP-PCR) assay to rapidly identify the expansion in 137/137 affected individuals from 16 independently reported FAME2 families worldwide (Fig. 1c, d, Table 1, Supplementary Table 1, Supplementary Data 1, Supplementary Fig. 2; Families 1, 3–6, 8–10, 12–16, 19, 20 and 22). Of the 24 individuals tested in these families that did not have a FAME diagnosis, two were positive for the ATTTC expansion; both were from younger generations and likely pre-symptomatic. We tested an additional 72 individuals (52 unrelated and 20 cases from six families with multiple affected individuals) with clinical similarity to FAME. Of these, 20/20 familial and 1/52 singleton cases were positive for an ATTTC expansion in *STARD7* (Table 1, Supplementary Fig. 2; Families 2, 7, 11, 17, 18 and 21 [singleton case]). The 52 unrelated subjects comprised 13 subjects with generalised epilepsy and tremor and 39 with myoclonic epilepsy with onset over the age of 19 years; 8/ 52 cases had a family history of epilepsy. Finally, within the families we tested, there were 13 individuals where the diagnosis of FAME was uncertain, usually due to a history of tremor with no other diagnostic features. Of these, 8/13 carried the ATTTC expansion. Two of the individuals with uncertain diagnosis that tested negative, were a mother and daughter pair from Family 1 (Supplementary Fig. 2a [red box] III-13 and IV-65) and

subsequent analyses with microsatellite markers showed that these individuals did not have the same haplotype as affected carriers of the ATTTC expansion (Supplementary Fig. 3). The ATTTC repeat expansion did not amplify in any of 28 control DNA samples extracted from unaffected individuals unrelated to FAME.

In all 158 individuals that tested positive for the ATTTC expansion, we observed that priming from ATTTT repeats was only successful from the telomeric end of the endogenous repeat and priming from ATTTC repeats was only possible from the centromeric end of the endogenous repeat. This suggested the structure of the pathogenic repeat in the context of the forward strand of chr2 was (AAATG)n[N](AAAAT)n, where (n) represents the unknown number of each repeat sequence.

**Long-read sequencing reveals the repeat structure.** The total numbers of repeats could not be determined by the RP-PCR assay, therefore we investigated some of these with long-read sequencing (Fig. 2). In one individual from the Australian-New Zealand family (Family 1: IV-98) a single molecule real-time (SMRT) read and a single Oxford Nanopore read were found that spanned the repeat. The SMRT read generated to 99% base accuracy by circular consensus calling was comprised of four subreads and contained 274 AAATG and 387 AAAAT repeats, without interruption from other sequences. The Oxford Nanopore read contained 345 AAATG and 390 AAAAT repeats with some interruptions, suggesting somatic variation of repeat sizes

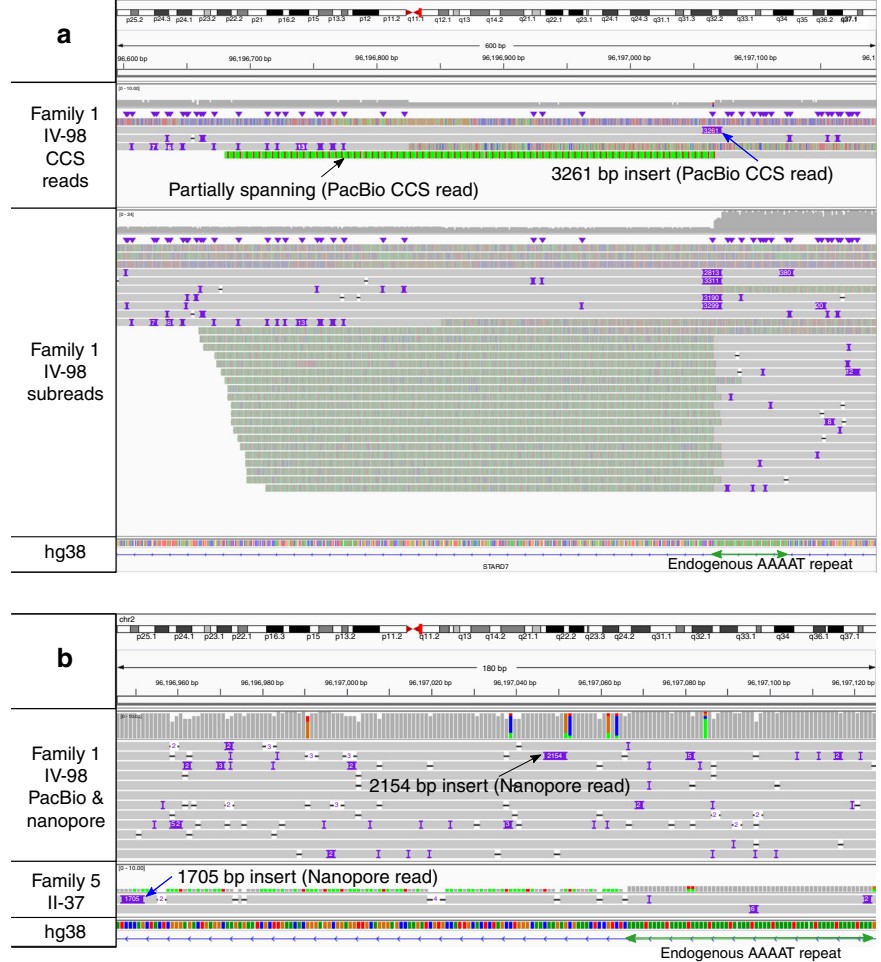

**Fig. 2** Long-read sequencing identifies the structure of the AAATG/AAAAT repeat expansion in intron one of *STARD7*. **a** Upper panel shows CCS reads from one member of Family 1 (IV-98) mapped to GRCh38. A read with a 3261 bp insert (blue arrow) which contains both AAATG and AAAAT sequences and flanking sequences that map to either side of the endogenous AAAAT repeat is present. Lower panel shows the component subreads mapped to the same region. **b** Top panel shows combined PacBio and nanopore reads mapped to hg38, following correction with Canu v1.7, with base pair mismatches in the reads masked for clarity. For Family 1 IV-98 (upper panel), a 2154 bp insert is shown (black arrow) on IGV; however, the read sequence contains a 3672 bp combined AAATG/AAAAT repeat insertion. Lower panel shows a nanopore read in one individual from Family 5 (II-37) with a 1705 bp insert on IGV (blue arrow), however the read contains a 4645 bp combined AAATG/AAAAT repeat insertion. Complete sequences for all reads that span the repeat expansion are included in Supplementary Data

may occur within the one individual. In a second individual (Family 5; III-37), a single Oxford Nanopore read spanned the expanded repeats with 588 AAATG and 340 AAAAT repeats; 4645 bp in total length. The natural variability in the length of the endogenous ATTTT repeat sequence meant that is was not feasible to use that sequence for mutation screening; however, the ATTTC repeat primer was diagnostic for FAME with a sensitivity of 100% in all families with linkage or suggestive linkage to chr2. This included two families with the previously identified *ADRA2B*; c.675_686delTGGTGGGGCTTTinsGTTTGGCAG; p. H225_L229delinsQ225_F_G_R228 variant strongly suggesting that allele is not causative (Table 1; Family 4 & 15)[17].

**Evidence of anticipation in a large FAME2 family.** In view of the discovery that FAME2 and FAME1 are caused by similar dynamic mutations of ATTTC repeats, and the demonstration of clinical anticipation in FAME1[11], we searched for evidence of anticipation in our pedigrees. We examined the median onset age of any relevant symptom, where available, for each generation in the Australian/New Zealand family (Family 1). We found evidence of anticipation; generation III had a median onset of

30 years (range 14–60 y, $n = 6$), in generation IV median onset was 17 years (8–50 y, $n = 30$) and the median onset in generation V was 12 years (4–19 y, $n = 16$). The remaining families were either too small or onset data were unavailable for anticipation to be robustly assessed.

**STARD7 transcript and protein abundance are not altered.** Reverse transcriptase, quantitative PCR using primer pairs spanning the repeat containing intron between exons one and two and a second pair spanning between exons three and four showed no significant differences in STARD7 transcript expression in patient-derived fibroblast cell lines (Fig. 3a). Protein abundance was also unaltered, confirmed by western blotting using an antibody to STARD7 protein that was previously validated using STARD7-knockout cell lines (Fig. 3b)[18]. RNA-Seq data from six patient-derived fibroblasts (four from Family 1 and two from Family 5) showed there was no significant difference in gene expression of STARD7 between affected and unaffected individuals along the entire length of the gene (Supplementary Fig. 4; $p = 0.838$; False Discovery Rate = 1). Reads containing ATTTC repeats were not present in the RNA-Seq data despite robust expression of

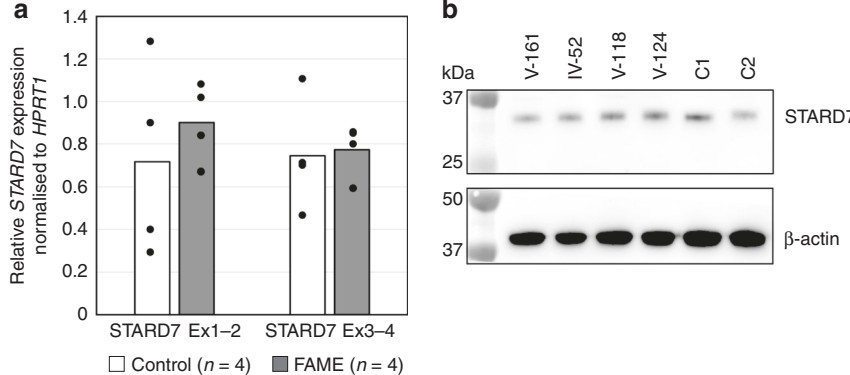

**Fig. 3** Expression of STARD7 is unaltered in patient-derived skin fibroblasts. **a** Graph shows average STARD7 expression by relative standard curve quantitative PCR (qPCR) normalised to HPRT1 expression in fibroblast cell lines from four control donors (white bars) and four affected male individuals from Family 1 (IV-52, V-118, V-124 and V-161; black bars). Individual data points overlay the each bar. Tests for significance were performed using Student's two-tailed $t$-test assuming unequal variances ($p = 0.50$ Exon 1–2; $p = 0.85$ Exon 3–4). **b** Western blot of STARD7 protein compared to β-actin on the same blot of fibroblasts from the same four individuals from Family 1 as assayed by qPCR and two male control donors (C1 and C2). Data underlying this figure are available in Source Data

STARD7. This is consistent with the observations from lymphoblastoid cell lines (LCLs) derived from individuals with FAME1, where no reads with repeats were found[11].

## Discussion

The pathogenic ATTTC insertion and expansion was always accompanied by the endogenous ATTTT pentanucleotide repeat in all cases of FAME2 that we describe here, replicating the findings in the cases of FAME with expansions in SAMD12, TNRC6A, RAPGEF2[10,11,19] and the report of a similar expansion in MARCH6 causing chr5-linked FAME[20]. The same observation also holds for spinocerebellar ataxia 37 (SCA37, OMIM: 615945), which is caused by the same repeat expansion in the first intron of DAB1[21]. For SCA37, it has been hypothesised that the thymidine to cytosine transition occurs after expansion of the endogenous ATTTT repeat to ~200 copies followed by further expansion of the mutant ATTTC sequence[22]. The ATTTT/ATTTC strand of the repeat is aligned with the direction of gene expression in all genes reported thus far, regardless of their chromosomal orientation. The mechanism of disease pathogenesis has been suggested to be RNA toxicity[21]. In zebrafish embryos, direct injection of RNA containing 58 copies of the AUUUC repeat was lethal or caused developmental defects in 81%, while the effect of injecting RNA containing 139 AUUUU repeats was not significantly different from controls[21]. Accumulation of AUUUC repeat containing RNA was observed in the brain of some individuals with FAME1, but we did not have access to similar biopsy tissue from individuals with FAME2[11]. While we found no significant change in expression of STARD7 in patient-derived cell lines, it is possible that expression of this gene is regulated differently in the non-proliferating cells of the brain. Profiling expression of all known genes implicated with pathogenic ATTTC dynamic mutations using gene expression data from the GTEX portal https://www.gtexportal.org[23] shows that DAB1 has high expression specifically in cerebellum while the five genes implicated in FAME thus far are more broadly expressed throughout the brain (Fig. 4). This difference in expression may partly explain the absence of epilepsy in individuals with SCA37.

STARD7 is a member of the START (StAR-related lipid transfer) domain-containing family of lipid transfer proteins with functions including intra-mitochondrial lipid transfer of phosphatidylcholine[24]. Previously, increased levels of choline have been detected by proton magnetic resonance spectroscopy ($^1$H-MRS) in the cerebellum of 11 individuals from three Italian families all shown here to have the ATTTC dynamic mutation[25] (Table 1). This observation may be peculiar to FAME2 families since the SAMD12, RAPGEF2, TNRC6A and MARCH6 genes do not have overlapping molecular functions.

In conclusion, we have identified the molecular basis of FAME2 is an inserted expanded ATTTC repeat in the first intron of the STARD7 gene, in 22 pedigrees with 266 affected individuals. The insertion segregates with disease status in 100% of individuals tested from families with linkage or suggestive linkage to chromosome 2 providing substantial genetic evidence that this mutation is causal in this syndrome. The FAME2 locus is the most frequently observed linked region for Caucasian individuals affected by this disorder whereas chromosome 8 thus far is limited to Asian individuals, therefore molecular genetic testing should take this into consideration if choosing to screen by RP-PCR. Identification of the gene and causative mutation for FAME2 opens the opportunity to explore the origins of the ATTTT/ATTTC expansion through a detailed comparison of the haplotypes and repeat structures of these individuals as has been done for SCA37[22]. There may be many additional undiagnosed individuals with a spectrum of FAME-related symptoms whose genetic causes may be due to ATTTC insertion and expansion at one of the FAME loci. This is especially likely in families that have multiple individuals with tremor and a low frequency of GTCS. As no preventive or curative treatments are currently available for FAME, these findings may have important therapeutic implications, including RNA-targeting treatments, such as antisense oligonucleotides or RNA-targeting Cas9 (RCas9)[26].

## Methods

**Ethics.** This study was approved by the Human Research Ethics Committees of the University of Melbourne and the University of Adelaide. Written, informed consent was obtained from all participants in the study.

**Whole-genome sequencing.** Adelaide: Human genomic DNA extracting from peripheral blood lymphocytes was prepared from two individuals in Family 1 (IV-3 and V-118) for sequencing using the TruSeq Nano DNA Library Preparation Kit (Illumina). Mapping of 150 bp, paired-end sequence reads to the UCSC hg19 build of the genome and calling of single nucleotide variants from whole-genome sequencing (WGS) data generated using an Illumina HiSeqX10 platform (Kinghorn Centre for Clinical Genomics, Sydney, Australia) was performed as previously described with the minor modification of using the Genome Analysis Toolkit (GATK) version 3.8 software[27,28]. Filtering of both coding and non-coding variants within the chr2 linkage interval shared between both individuals under a dominant model and absent from the gnomAD variant database[29] at a frequency >0.001 was performed using the bcftools isec command from htslib v1.9. Single nucleotide variants and indels were annotated with ANNOVAR[30]. Reads

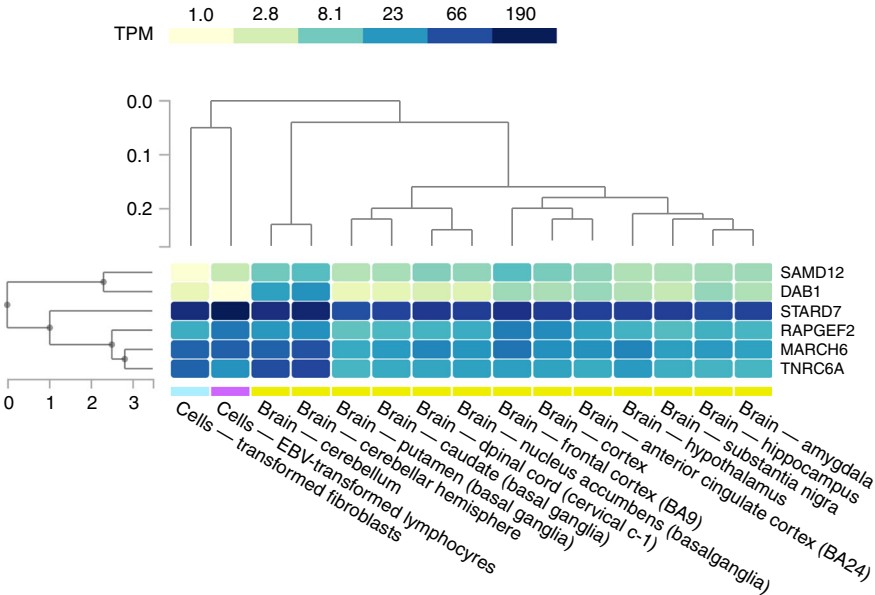

**Fig. 4** Expression patterns of ATTTC repeat genes in brain, skin fibroblast and lymphoblastoid cell lines. The heatmap shows relative gene expression expressed as transcripts per kilobase per million mapped reads (TPM) based on the colour scale as shown. Data and image downloaded from the GTEx Portal https://www.gtexportal.org

containing the expanded repeat were visualised using the Integrative Genomics Viewer (IGV) v2.4.5 with soft-clipped reads unmasked[31].

Rome: WGS library was prepared from the genomic DNA of the individual (PM195; Family 4) by using TruSeq DNA PCR-Free KIT (Illumina, San Diego, CA, USA) and sequenced 150 bp paired-end reads on an Illumina HiSeq producing 470,174,247 fragments, corresponding to about 39X coverage after mapping and removal of duplicated reads. Reads were quality filtered and aligned to the reference human genome sequence (GRCh38/hg38) with BWA-MEM v.0.7.15[32]. Resulting BAM files underwent local realignment around insertion-deletion sites, duplicate marking and recalibration steps with GATK v3.8[28]. Variant calling was performed with HaplotypeCaller v3.8 with standard parameters, and output VCF files were recalibrated with VariantRecalibrator from GATK v3.8. Genomic variant annotation was carried out with VarSeq v1.4.7 (Golden Helix, Inc., Bozeman, MT, www.goldenhelix.com) and only variants with a minimum read depth of 5X were included in the downstream analysis. Thereafter, only variants in the pericentromeric region of interest of chr2 (chr2: 91,800,000–106,700,000) were considered.

Prioritisation of variants of potential interest was carried out through three distinct analyses. For the first analysis, all variants reported to be pathogenic or potentially pathogenic in the clinical databases of ClinVar, HGMD Professional v2017.2 and/or Centogene CentoMD v4.1 were retained. For the second analysis, we focused on variants in exonic regions without a reported clinical annotation. We excluded variants with a population frequency above 1% in the databases of 1000 Genomes Project, National Heart, Lung and Blood Institute (NHLBI, https://www.nhlbi.nih.gov/) Exome Sequencing Project (ESP, http://evs.gs.washington.edu/), ExAC (Exome Aggregation Consortium, http://exac.broadinstitute.org/) and gnomAD (The Genome Aggregation Database, https://gnomad.broadinstitute.org/), along with variants recorded in the Personal Genomics internal database. We retained all the non-synonymous variants predicted to alter the protein structure or function by at least three of the following in silico prediction tools: Mutation Taster, SIFT, Polyphen-2, MutationAssessor and FATHMM. For the third analysis, we prioritised the variants outside exonic regions by considering rare variants (frequency below 1% in frequency population databases, including the Personal Genomics internal database) and with a predicted significant effect on the protein structure or function by at least three of the in silico prediction tools. Variants were then prioritised by considering their presence in regulatory regions as reported in the ENCODE database (https://www.encodeproject.org/). The manual inspection of the BAM files, by using Integrative Genomics Viewer (IGV), allowed us to evaluate the coverage of the variants and the quality of the aligned reads.

The identification of putative genomic expansions, structural variants or copy number variations was carried out by using Lumpy v0.2.13[33] and Manta v1.2.2[34] software. The ExpansionHunter tool v2.5.3[14] was adopted to estimate the size of potential repetitions of short unit sequences.

**Long-read sequencing**. DNA was extracted for all long-read sequencing protocols using the QIAsymphony system from skin fibroblasts (passage 6) cultured in Dulbecco's modified Eagle's Medium (DMEM; Life Technologies) with 10% fetal calf serum. Pacific Biosciences (PacBio) single molecule real-time (SMRT)

sequencing data were obtained in two batches: In the first batch, two Australian FAME2 carriers (Family 1: IV-44 and IV-98) were sequenced with two flow cells per sample. Resulting bam files were converted to fastq using the SMRT Link software v5.1.0 *bam2fastq* program. Resulting fastq files were either mapped directly to the human genome hg38 build using NGM-LR[35] with structural variants called by Sniffles[35] or used as input for de novo assembly with Canu v1.7. In the second batch, a single sample (Family 1: IV-98) was sequenced. DNA fragment sizes were determined with the Femto Pulse capillary electrophoresis system (Agilent Technologies, Santa Clara, CA). DNA fragments of size greater than 6 kb were selected with BluePippin (Sage Science, Beverly, MA) pulsed field gel electrophoresis system. Sequencing was carried out for 20 h per SMRT cell on the Sequel system with Binding Kit 3.0 (PacBio, 101–500–400) and Sequencing Kit 3.0 (PacBio 101–427–800). Circular consensus calling was performed using CCS 3.2.1 software. Reads were mapped to the GRCh38 build of the human genome using *pbmm2* with "-c 0 -L 0.01" for CCS reads and "-c 0 -L 0.1" for subreads.

Oxford nanopore data were obtained for DNA samples extracted from fibroblasts from two individuals from Family 1, as described above, and two from Family 5 (II-37 and IV-29 Fig. S2e). For each of the four participant samples, 3 µg of DNA was prepared for Oxford Nanopore 1D genomic sequencing by ligation using the SQK-LSK108 kit and was run on a FLO-MIN106 flow cell for 48 h. Basecalling was performed on MinKNOW 18.01.6 with MinKNOW Core 1.11.5 and Albacore v2.1. Data were either mapped with NGM-LR or assembled with Canu v1.7 as described below, using suggested settings for nanopore sequencing reads.

De novo whole-genome assembly of one individual with input of both PacBio and nanopore sequencing from one individual from Family 1 was carried out using the Canu v1.7 assembler with default starting parameters for a genome size of 3.6 Gbp. Recalibrated reads from Canu were mapped to the hg38 build of the human genome using NGM-LR as described above.

**Repeat expansion analysis**. WGS was performed for two affected individuals from Family 1 on the Illumina HiSeq X10 platform, one individual from Family 3 as described above, and three affected individuals from Family 19 on the Illumina HiSeq platform. A cohort of 69 individuals without FAME were used for comparison, with 150 bp paired-end sequencing performed on the Illumina HiSeq X platform (Kinghorn Centre for Clinical Genomics, Sydney, Australia). Library preparation for 53 of the samples used the Illumina TruSeq Nano DNA HT Library Preparation Kit; the other 16 samples used KAPA Hyper Prep Kit PCR-free library preparation.

Reads were aligned to the hg19 reference genome with BWA-MEM v0.7.17-r1188[32], then duplicate marking, local realignment and recalibration were performed with GATK v4.0.3.0[28]. Repeat expansion analysis targeting two FAME2 loci, the ATTTT repeat and predicted ATTTC insertion in STARD7, was performed using ExpansionHunter v2.5.5[14] and exSTRa v0.88.3 with Bio-STR-exSTRa v1.0.1[15]. Custom files defining the FAME2-AAAAT and FAME2-AAATG repeat loci were created for ExpansionHunter (below) and exSTRa (Supplementary Table 2).

```
{
  "OffTargetRegions": [
  "chr3:151086374151086421",
  "chr4:2171641221716717",
  "chr6:4867170448671855",
  "chr8:119379052119379154",
  "chr9:113463975113464205"],
  "RepeatId": "FAME2-AAAAT",
  "RepeatUnit": "AAAAT",
  "TargetRegion": "chr2:9686280596862859"
}
{
  "OffTargetRegions": [
  "chr3:151086374151086548",
  "chr4:2171641221716717",
  "chr6:4867170448671855",
  "chr8:119379052119379154",
  "chr9:113463975113464205"],
  "RepeatId": "FAME2-AAATG",
  "RepeatUnit": "AAATG",
  "TargetRegion": "chr2:9686282596862826"
}
```

Supplementary Figure 1 shows the repeat sizes predicted by ExpansionHunter and empirical cumulative distribution function of repeated bases from exSTRa for the two FAME2 loci. Significance testing was performed using the exSTRa tsum_test function with 100,000 permutations in case-control mode comparing each affected individual with FAME to the 69 unaffected individuals without FAME. All FAME2 carriers were significant outliers for the FAME2-AAATG locus ($p < 0.0001$ for all individuals) while only four samples were significant outliers ($p < 0.05$) for the FAME2-AAAAT locus.

**RNA-Seq.** Total RNA was extracted from patient-derived primary skin fibroblasts of four Australian/New Zealand FAME, two Italian FAME and four age-matched controls using QIAGEN RNeasy kits, as per the manufacturer's protocol. Library preparation and RNA-Seq were performed as a service by the UCLA Neuroscience Genomics Core Facility. The TruSeq v2 kit (Illumina) was used to generate un-stranded libraries with 150-bp mean fragment sizes and 50-bp paired-end sequencing performed using the HiSeq2500 (Illumina). Sequence data were mapped to the GRCh38 build of the human genome using *HISAT2 v2.1.0*[36]. Read counts were generated with *StringTie v1.3.3*[36]. Differential expression between FAME and control samples was determined using the exact test from the *edgeR v3.26.5* package in *R v3.6.0*[37]. Differentially expressed genes were filtered to false discovery rate (FDR) < 0.05 and log base 2-fold change (LFC) > = 1 or < = −1.

**Quantitative PCR.** RNA was extracted from four patient-derived primary skin fibroblast cell lines from Family 1 and four control fibroblast cell lines from adult donors not affected by FAME as described above under *RNA-Seq*. cDNA were generated from 1 μg of total RNA using the iScript reverse transcription kit (Bio-Rad, Gladesville, NSW, Australia; cat# 1708891), according to the manufacturer's protocol.

Quantification of differentially expressed transcripts was performed with the relative standard curve method using SYBR green fluorescence intensity for detection. Products were amplified in 1 × iTaq Universal SYBR Green supermix (Bio-Rad; cat# 1725121) with primers at 1μM final concentration. Each sample and standard was amplified with three technical replicates on an Applied Biosystems StepOnePlus. Expression values were determined relative to a dilution curve of a cDNA standard made from pooled control fibroblast cDNA. Specificity of products was determined by melt curve analysis at the conclusion of each run. Expression values of each gene were normalised to *HPRT1* expression values from the same sample.

**Western blotting.** Fibroblasts were cultured as described in Supplementary methods then lysed with lysis buffer (150 mM NaCl, 1% Triton X-100, 1 mM EDTA, 0.25% Sodium deoxycholate, 50 mM Tris. Added protease inhibitor, 50 mM NaF and 0.1 mM Na$_3$VO$_4$). Extracts were separated by 4–12% polyacrylamide gel and transferred to nitrocellulose membrane by electroblotting. STARD7 was detected with rabbit polyclonal anti-human/mouse/rat STARD7 (Proteintech cat# 15689–1-AP) at 1:500 dilution followed by anti-rabbit IgG conjugated to horse-radish peroxidase (HRP) at 1:2000 (Dako cat# P0448). Enhanced chemilumines-cent detection (Bio-Rad cat# 1705061) was visualised with the chemidoc detection system (Bio-Rad). Full blots are available in the Source Data file.

**PCR amplification and sequencing of repeats (Rome).** Pentanucleotide repeats were analysed in duplicate by long-range PCR with Expand Long Template PCR System (Roche) according to the manufacturer's recommendation. Some 200 ng genomic DNA were amplified with primers STARD7F and STARD7R (300 nM), dNTP (350 μM) buffer 1 (1×) Enzyme 0.5 U (×50 μl reaction). After 2 min of initial denaturation at 94 °C, DNA samples underwent 10 cycles of amplification (dena-turation 94 °C for 10 s, annealing 56 °C for 30 s, elongation 68 °C 3 min) followed by an additional 20 cycles (94 °C for 15 s, annealing 56 °C for 30 s, elongation 68 °C 45 s + 20 s each cycle elongation for each successive cycle). PCR products were separated by electrophoresis on 1% agarose gel. DNA was extracted from the agarose gel slice and the number of repeat units was determined by Sanger sequencing (Eurofins Genomics Sequencing Service).

**Repeat-primed PCR.** Primers for both Adelaide and Rome are shown in Sup-plementary Table 3.

Adelaide: Reaction mixes included 100 ng genomic DNA, 0.5 μM FAM-labelled locus specific (RP-PCR-FAME2-P1 or P2) and RP-PCR-P3 primers, and 0.05 μM repeat specific primer (one of RP-PCR-FAME2–4.5 to 4.8) with Expand Long Template polymerase (Roche, cat# 25524324) or Taq polymerase (Roche, cat# 18697220). The initial RP-PCR step was at 95 °C for 5 min followed by 10 cycles (95 °C for 30 s, 48 °C + 1.0 °C each cycle for 45 s and 65 °C + 1.0 °C each cycle for 5 min) continuing to 30 cycles (95 °C for 30 s, 58 °C for 1 min and 72 °C for 5 min) and ending with 72 °C for 7 min. Fragment analysis was performed on the RP-PCR products with an ABI3730 DNA analyser.

Rome: The pentanucleotide repeat sequence in *STARD7* gene was amplified by ATTTT and ATTTC RP-PCR with the following primers: STARD7R* 5′FAM-labelled (locus specific primer), RP-PCR-STARD7-P3 (generic primer) and RP-PCR-STARD7-P4 primers specific for the short pentanucleotide repeat (ATTTT) and for the possible expanded (ATTTC) repeat or possible (ATTTC) repeat interruption. PCR was performed with 100 ng DNA, 1.5 mM MgCl$_2$, 200 μM dNTP, 0.4 μM locus specific primer, 0.4 μM generic primer, 0.2 μM repeat primer, 2.5 U Polymed Taq in 25 μl volume. The initial PCR step was at 94 °C for 15 min followed by 35 cycles (94 °C for 45 s, 60 °C for 30 s and 72 °C for 2 min) and 72 °C elongation for 30 min. Capillary electrophoresis was performed on ABI310 GEN ANALYZER (Applied Biosystems).

**Reporting summary.** Further information on research design is available in the Nature Research Reporting Summary linked to this article.

## Data availability

Source data for Figs. 1a, 3a, 3b, Supplementary Figs. 1a, b and 4b are provided in the Source Data files of this manuscript. RNA-Seq data are available from the NCBI BioProject PRJNA563467. Whole-genome sequencing data are available from the corresponding author on request, subject to human research ethics approval and patient consent.

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

## Acknowledgements
We wish to thank the many families involved in this study. We thank Dr. Tessa Mattiske, Dr. Mark Holloway and (Andy) Hung Nguyen for technical assistance. Dr. Joel Geoghegan and Dr. Andreas Schreiber for assistance with PacBio sequencing. We wish to acknowledge the following sources of funding: NHMRC (Jozef Gecz, Ingrid Scheffer Sam Berkovic), Women's and Children's Hospital Research Foundation (Mark Corbett, Jozef Gecz), Muir Maxwell Trust and Epilepsy Society (Simona Balestrini, Sanjay M. Sisodiya), The European Fund for Regional Development from the European Union (grant 01492947) and the province of Friesland, Dystonia Medical Research Foundation, Stichting Wetenschapsfonds Dystonie Vereniging, Fonds Psychische Gezondheid, Phelps Stichting, from Ipsen & Allergan Farmaceutics, Merz, and Actelion (Marina A.J. Tijssen). The Italian Ministry of Health (grant GR2013–02356227) and Istituto Superiore di Sanità, Italy, (grant PGR00229-PGR00919 and Farmindustria) Undiagnosed Disease Network Italy, (Francesco Brancati). The Fondation maladies rares, University Hospital Essen (Christel Depienne). This work was partly done at NIHR University College London Hospitals Biomedical Research Centre, which receives a proportion of funding from the UK Department of Health's NIHR Biomedical Research Centres funding scheme.

## Author contributions
M.A.C.: Coordination of the FAME consortium, drafted the manuscript, bioinformatics (WGS and RNA-seq), designed molecular assays, qPCR. T.K.: RP-PCR assays, western blots, Sanger sequencing (Adelaide), contributed to writing the manuscript. Liana Veneziano: RP-PCR assays and Sanger sequencing (Rome), contributed to writing the manuscript. M.F.B.: Bioinformatic analysis of WGS data for repeat expansions, contributed to writing the manuscript. R.F.: Genetic analysis of French, German and Dutch FAME families, contributed to writing the manuscript. A.L.S.: Recruitment of families and phenotyping, contributed to writing the manuscript. A.C.: Recruitment of families, collection of blood samples, collection of clinical and electrophysiological data, collection of DNA and phenotyping, contributed to writing the manuscript. L.L.: Recruitment and phenotyping, collection of clinical and electrophysiological data, collection of DNA, contributed to writing the manuscript. S. Franceschetti.: Recruitment of families and phenotyping, contributed to writing the manuscript. A.S.: Clinical study and recruitment, contributed to writing the manuscript. A.W.: PacBio data analysis, contributed to writing the manuscript. D.M.: Genetic study of ADCME-FAME2 family, contributed to writing the manuscript. M.P.: Molecular analysis, linkage analysis, contributed to writing the manuscript. S.K.: Screening of FAME families for expansion, contributed to writing the manuscript. M.D.: Genome sequencing analysis (Italy), contributed to writing the manuscript. Rachel Straussberg: Recruitment of one family and phenotyping, contributed to writing the manuscript. Luciano Xumerle: Genome sequencing and data analysis (Italy), contributed to writing the manuscript. B.R.: Recruitment of families and phenotyping, contributed to writing the manuscript. D.C.: Recruitment, phenotyping and detail analysis of Family 1, contributed to writing the manuscript. A-F.v.R.: Recruitment, phenotyping, collection of clinical and electrophysiological data, collection of DNA, contributed to writing the manuscript. A.C.: Repeat-primed PCR (Adelaide), contributed to writing the manuscript. R. Catford.: Molecular analysis (Adelaide), contributed to writing the manuscript. F. Bisulli.: Recruitment of families and phenotyping, contributed to writing the manuscript. S. Chakraborty.: PacBio sample preparation and sequencing, contributed to writing the manuscript. S.B.: DNA/RNA extraction, DNA genotyping (microsatellites/SNPs), fibroblast cultures, contributed to writing the manuscript. P.T.: Recruitment of the families, clinical and neurophysiological phenotyping, contributed to writing the manuscript. K.B.: Nanopore sample preparation and sequencing, contributed to writing the manuscript. S. Carswell.: Nanopore sample preparation and sequencing, contributed to writing the manuscript. M.S.: Nanopore sample preparation and sequencing, contributed to writing the manuscript. A.B.: Clinical characterisation, contributed to writing the manuscript. R. Carroll.: Culturing of FAME fibroblasts and qPCR for RNA-Seq, contributed to writing the manuscript. A.G.: RNA-seq and WGS analyses

(Adelaide), contributed to writing the manuscript. K.F.: Molecular analysis (Adelaide), contributed to writing the manuscript. I.B.: Recruitment of a family and phenotyping, contributed to writing the manuscript. M.I.: Collection of DNAs and linkage analysis, contributed to writing the manuscript. C.D.B.: Recruitment of one family, contributed to writing the manuscript. S.S.: Recruitment of families and phenotyping, contributed to writing the manuscript. J.B.: Bioinformatic analysis (Essen), contributed to writing the manuscript. B.K.: Genetic analyses: genotyping, linkage, sequencing, bioinformatic analysis, contributed to writing the manuscript. C.N.: Genetic analyses: genotyping, linkage, sequencing, collection of FAME samples, contributed to writing the manuscript. S. Forlani.: DNA/RNA extraction, cell culture (lymphoblasts, fibroblasts), coordination of FAME samples in France, contributed to writing the manuscript. G.R.: Recruitment of families and phenotyping, contributed to writing the manuscript. E.L.: Initiation of FAME project in France, collection of FAME families, contributed to writing the manuscript. E.H.: Recruitment of families and phenotyping, contributed to writing the manuscript. P.L.: Initiation of FAME project in France, recruitment and phenotyping of FAME families, contributed to writing the manuscript. S.B.: Recruitment of a family and phenotyping, contributed to writing the manuscript. J.W.S.: Recruitment of a family and phenotyping, contributed to writing the manuscript. Z.A.: Collection of DNAs and linkage analysis, contributed to writing the manuscript. I.H.: Phenotyping, collection of clinical data, linkage analysis, contributed to writing the manuscript. H.I.: Provided technical and intellectual advice, contributed to writing the manuscript. S.T.: Provided technical and intellectual advice, contributed to writing the manuscript. S.M.S.: Recruitment of a family and phenotyping, contributed to writing the manuscript. G.C.: Genetic study of ADCME-FAME2 family, contributed to writing the manuscript. L.G.S.: Recruitment of individuals and phenotyping, collection of clinical and electro-physiological data, collection of DNA, contributed to writing the manuscript. R.v.C.: Recruitment and phenotyping of South African family, contributed to writing the manuscript. M.AJ.T.: Recruitment of individuals and families, phenotyping, collection of clinical and electrophysiological data, collection of DNA, contributed to writing the manuscript. K.M.K.: Phenotyping, collection of clinical and electrophysiological data, linkage analysis, contributed to writing the manuscript. A..v.d.M.: Recruitment and genetic study of one family, contributed to writing the manuscript. F.Z.: Collection of DNAs and linkage analysis, contributed to writing the manuscript. R.G.: Clinical, neu-rophysiological and genetic study of ADCME-FAME2 family, contributed to writing the manuscript. S.F.B.: Co-led family study, phenotyping and analysis of syndrome, collection of DNA, identification of families and sporadic cases, contributed to writing the manuscript. T.P.: Sample selection, linkage analysis, fibroblast cultures, exome sequencing, contributed to writing the manuscript. Laura Canafoglia: Recruitment of the families and phenotyping, contributed to writing the manuscript. M.B.: Bioinformatic analysis of WGS data for repeat expansions, contributed to writing the manuscript. P.S.: Recruitment of the families and phenotyping, collection of clinical and electro-physiological data, contributed to writing the manuscript. I.E.S.: Co-led family study, phenotyping and analysis of syndrome, collection of DNA, identification of families and sporadic cases, contributed to writing the manuscript. F. Brancati.: Genetic studies, coordination of the work to identify the genetic cause in a FAME2 Italian family, contributed to writing the manuscript. C.D.: PI and coordinator of FAME projects in France, the Netherlands and Germany. Genetic analyses (linkage, WGS, RNA-seq, test or confirmation of expansion) in 10 FAME families, contributed to writing the manuscript. J.G.: Co-lead investigator (Australia), Coordinated and designed molecular and genetic assays, designed the project, contributed to writing the manuscript.

## Competing interests

A.W. and S. Chakraborty. are employees and shareholders of Pacific Biosciences. There are no other competing interests to declare.

## Additional information

Mark A. Corbett[1], Thessa Kroes[1], Liana Veneziano[2], Mark F. Bennett[3,4,5], Rahel Florian[6], Amy L. Schneider[5], Antonietta Coppola[7], Laura Licchetta[8,9], Silvana Franceschetti[10,11], Antonio Suppa[12,13], Aaron Wenger[14], Davide Mei[15], Manuela Pendziwiat[16], Sabine Kaya[6], Massimo Delledonne[17], Rachel Straussberg[18,19], Luciano Xumerle[20], Brigid Regan[5], Douglas Crompton[5,21], Anne-Fleur van Rootselaar[22], Anthony Correll[23], Rachael Catford[23], Francesca Bisulli[8,9], Shreyasee Chakraborty[14], Sara Baldassari[8], Paolo Tinuper[8,9], Kirston Barton[24], Shaun Carswell[24], Martin Smith[24,25], Alfredo Berardelli[12,13], Renee Carroll[1], Alison Gardner[1], Kathryn L. Friend[23], Ilan Blatt[26], Michele Iacomino[27], Carlo Di Bonaventura[12], Salvatore Striano[28], Julien Buratti[29], Boris Keren[29], Caroline Nava[30], Sylvie Forlani[30], Gabrielle Rudolf[31,32,33,34,35], Edouard Hirsch[34], Eric Leguern[29,30], Pierre Labauge[36], Simona Balestrini[37,38], Josemir W. Sander[37,38], Zaid Afawi[19], Ingo Helbig[39,16], Hiroyuki Ishiura[40], Shoji Tsuji[40,41,42], Sanjay M. Sisodiya[37,38], Giorgio Casari[43], Lynette G. Sadleir[44], Riaan van Coller[45], Marina A.J. Tijssen[46], Karl Martin Klein[47,48,49], Arn M.J.M. van den Maagdenberg[50], Federico Zara[27], Renzo Guerrini[15], Samuel F. Berkovic[5], Tommaso Pippucci[51], Laura Canafoglia[10,11],

Melanie Bahlo [3,4], Pasquale Striano [52,53], Ingrid E. Scheffer [5,54], Francesco Brancati [2,55,56], Christel Depienne [6,31,35] & Jozef Gecz [1,57]

[1]Adelaide Medical School and Robinson Research Institute, University of Adelaide, Adelaide 5005 SA, Australia. [2]Institute of Translational Pharmacology, National Research Council, Rome, Italy. [3]Population Health and Immunity Division, the Walter and Eliza Hall Institute of Medical Research, Parkville 3052 VIC, Australia. [4]Department of Medical Biology, the University of Melbourne, Melbourne 3010 VIC, Australia. [5]Epilepsy Research Centre, Department of Medicine, University of Melbourne, Austin Health, Heidelberg 3084 VIC, Australia. [6]Institut für Humangenetik, Universitätsklinikum Essen, Universität Duisburg-Essen, Essen, Germany. [7]Department of Neuroscience, Reproductive and Odontostomatological Sciences, Federico II University, Napoli, Italy. [8]IRCCS Istituto delle Scienze Neurologiche di Bologna, Bologna, Italy. [9]Department of Biomedical and Neuromotor Sciences, University of Bologna, Bologna, Italy. [10]Neurophysiopathology, Fondazione IRCCS Istituto Neurologico Carlo Besta, Milan, Italy. [11]Member of the European Reference Network on Rare and Complex epilepsies, ERN EpiCARE, London, UK. [12]Department of Human Neurosciences, Sapienza University of Rome, Viale dell'Università, 30, 00185 Rome, Italy. [13]IRCCS Neuromed, Pozzilli, IS, Italy. [14]Pacific Biosciences, Menlo Park, CA, USA. [15]Neuroscience and Neurogenetics Department, Meyer Children's Hospital, Florence, Italy. [16]Department of Neuropediatrics, University Medical Center Schleswig-Holstein, Christian-Albrechts University, Kiel, Germany. [17]Department of Biotechnology, University of Verona, Strada le Grazie 15, 37134 Verona, Italy. [18]Institute of Pediatric Neurology, Schneider Children's Medical Center of Israel, Petah Tikva, Israel. [19]Tel Aviv University Medical School, 69978 Tel Aviv, Israel. [20]Personal Genomics, Strada le Grazie 15, 37134 Verona, Italy. [21]Department of Neurology, Northern Health, Melbourne, VIC, Australia. [22]Amsterdam UMC, University of Amsterdam, Department of Neurology and Clinical Neurophysiology, Amsterdam Neuroscience, Amsterdam, The Netherlands. [23]Genetics and Molecular Pathology, SA Pathology, Adelaide, SA, Australia. [24]Kinghorn Centre for Clinical Genomics, Garvan Institute for Medical Research, Darlinghurst, NSW 2010, Australia. [25]St-Vincent's Clinical School, Faulty of Medicine, UNSW Sydney, Darlinghurst, NSW 2010, Australia. [26]Department of Neurology, Sheba Medical Center, Tel Hashomer, Israel. [27]Laboratory of Neurogenetics, IRCCS Istituto "G. Gaslini", Genova, Italy. [28]Department of Neurology, Federico II University, Napoli, Italy. [29]AP-HP, Hôpital Pitié-Salpêtrière, Département de Génétique, F-75013 Paris, France. [30]INSERM, U 1127, CNRS UMR 7225, Sorbonne Universités, UPMC Univ Paris 06 UMR S 1127, Institut du Cerveau et de la Moelle épinière, ICM, F-75013 Paris, France. [31]Institut de Génétique et de Biologie Moléculaire et Cellulaire, Illkirch, France. [32]Institut National de la Santé et de la Recherche Médicale, U1258 Illkirch, France. [33]Université de Strasbourg, Illkirch, France. [34]Department of Neurology, Strasbourg University Hospital, Strasbourg, France. [35]Centre National de la Recherche Scientifique, U7104 Illkirch, France. [36]MS Unit, Montpellier University Hospital, Montpellier, France. [37]Department of Clinical and Experimental Epilepsy, UCL Queen Square Institute of Neurology, London WC1N 3BG, UK. [38]Chalfont Centre for Epilepsy, Chalfont St Peter SL9 0RJ, UK. [39]Division of Neurology Children's Hospital of Philadelphia, Philadelphia, PA, USA. [40]Department of Neurology, the University of Tokyo Hospital, Tokyo, Japan. [41]Medical Genome Center, the University of Tokyo Hospital, Tokyo, Japan. [42]International University of Health and Welfare, Chiba, Japan. [43]TIGEM - Telethon Institute of Genetics and Medicine, Naples, and San Raffaele University, Milan, Italy. [44]Department of Paediatrics and Child Health, University of Otago, Wellington, Wellington, New Zealand. [45]University of Pretoria, Pretoria, South Africa. [46]Department of Neurology, University of Groningen, Groningen, The Netherlands. [47]Department of Neurology, Epilepsy Center Frankfurt Rhine-Main, Goethe University, Frankfurt am Main, Frankfurt, Germany. [48]Department of Neurology, Epilepsy Center Hessen,  Philipps University, Marburg, Marburg, Germany. [49]Departments of Clinical Neurosciences, Medical Genetics and Community Health Sciences, Hotchkiss Brain Institute & Alberta Children's Hospital Research Institute, Cumming School of Medicine, University of Calgary, Calgary, AB, Canada. [50]Departments of Human Genetics & Neurology, Leiden University Medical Centre, Leiden, The Netherlands. [51]Medical Genetics Unit, Sant'Orsola-Malpighi University Hospital, Bologna, Italy. [52]Pediatric Neurology and Muscular Diseases Unit, IRCCS Istituto "G. Gaslini", Genova, Italy. [53]Department of Neurosciences, Rehabilitation, Ophthalmology, Genetics, Maternal and Child Health, University of Genoa, Genova, Italy. [54]Royal Children's Hospital, Murdoch Children's Research Institute and Florey Institute, Melbourne, VIC, Australia. [55]Medical Genetics, Department of Life, Health and Environmental Sciences, University of L'Aquila, L'Aquila, Italy. [56]Laboratory of Molecular and Cell Biology, Istituto Dermopatico dell'Immacolata, IDI-IRCCS, Rome, Italy. [57]South Australian Health and Medical Research Institute, Adelaide 5000 SA, Australia.

