## [Peer Review File · Nature Communications]

Reviewers' Comments:

Reviewer #1:

Remarks to the Author:

In this manuscript by Gecz et al, they describe a new complex ATTTT/ATTTC pentanucleotide repeat expansion in intron 1 of STARD7 in a series of families with FAME. This is an important finding and builds on recent work demonstrating ATTTC repeat expansions at three other loci in FAME/BAFME families. Overall, this finding is important and the work is well done. The evidence supporting this repeat as the causative agent is compelling in light of the other recently published work with a few minor issues noted below. However, the manuscript itself requires some editing, with inclusion and expansion of some of the data already collected into the main body of the manuscript. In addition, while an expansive evaluation of how the repeat causes disease is beyond the scope of this manuscript, a few basic experiments should be done to better characterize how this repeat might cause disease.

Issues:

1) The two "false negative" cases need more characterization, as these pose the greatest potential challenge to these ATTTC repeats being the causative element. A few questions related to these cases:

- a. Do these cases carry the same disease associated haplotype region as the repeat that runs in their families or are they negative for the disease haplotype independent of their repeat findings?
- b. Are either of these cases obligate causative gene carriers?
- c. What is the evidence that they were phenocopies? Did they have the same EEG findings or just a vague history of myoclonus?
- d. If they also had epilepsy, was there any other causative gene in these cases (and not in other cases) that might reveal the underlying cause of their diseases to allow us to more easily discount them?

2) The abstract suggests that RNA-seq analysis revealed involvement of protein quality control pathways (the preferred term for the autophagy and ubiquitin proteasome system, rather than "Cellular waste collection"). However, there is only a reference to three supplemental tables to justify this statement and these tables are difficult to negotiate. If it is important enough to be in the abstract and discussed in the text, then it would be more appropriate for at least a summary of this RNA-seq data and GO analysis to be included within the main body of the manuscript- likely as part of a new figure.

3) If you want to propose that there are alterations in protein quality control pathways, then this will require more robust analysis than what has been done so far. Specifically, the Gene ontology analysis needs validation of the RNA-seq data by RT-PCR and (ideally) protein levels by immunoblot for key factors driving the GO analysis findings.

4) Mature STARD7 mRNA levels should be independently confirmed by RT-PCR using primers that cross exon-exon junctions to determine the abundance of the mature mRNA.

5) The lack of repeat sequences within the RNA-seq dataset does not mean that the RNA is not accumulating as a) If they used oligodT primers for cDNA generation then they will miss any intronic sequences that were not included in the mature mRNA b) Repeat RNAs are often retained in the nucleus, so if their RNA isolation protocol included a spin that eliminated the nucleus prior to RNA isolation, it might be missed and c) the repeats may simply not be readable by RNA-seq given their size and repetitive highly AT-rich nature. There are no controls shown to demonstrate that they could even detect this RNA if it was present. Is the surrounding intronic mRNA sequence enriched/observed? Some evaluation of whether the repeat containing intron is retained (either within the RNA-seq data or by using intron/exon junction primer sets or with RNA foci determination by smFISH) in patient cells would be helpful and would provide some information about how the repeat might lead to disease.

6) The STARD7 protein levels should be measured in patient cells compared to controls as a second measure for influence of the repeat on this gene.

Minor points:

- 1) It is unclear why there is only one figure but a huge number of supplemental figures, many of

which would be better in the main body of the manuscript. Figures S3, S4, and S5 contain information that will be of interest to the general reader, as would some visual representation of the data from tables S3-S6. This would make it easier for the readers to negotiate the manuscript and understand the findings and studies done.

2) Is the repeat on a single haplotype across all of the families?

3) Is there any evidence of anticipation in these families in terms of age of onset?

Reviewer #2:

Remarks to the Author:

Corbett et al identify an intronic expansion mutation in STARD7 as the cause of familial adult myoclonic epilepsy type 2 (FAME2; linked to chromosome 2).

The researchers analyzed WGS data from 6 individuals from 3 families. They searched for TTTTA and TTTCA expansions within the known linkage region of chromosome 2 using ExpansionHunter and exSTRa. They identified an expansion of the endogenous ATTTT as well as an insertion of an expanded ATTTC within the STARD7 intron. They then performed an assay to screen 160 individuals from 22 families linked to FAME2 locus. They showed the presence of the insertion expansion (ATTTC) in 158/160 and it was not present in controls. RNA studies showed that STARD7 was expressed similarly among 6 cases and controls. RNA-Seq showed differential expression of several hundred genes across the genome. Genes involved in the metabolism of waste products appeared to be enriched.

The manuscript presents highly convincing evidence that the STARD7 expansion is responsible for FAME2. This is a robust and novel finding. The quality of the writing is excellent and the methods are sound. Results are interpreted appropriately. The authors are leaders in the field of epilepsy genetics.

Comments:

1-Family 4 and 15 were previously published as having FAME2 due to mutation in ADRA2B. While I see that they comment on this variant in their Table, can the authors comment, in the text, that this molecular diagnosis was incorrect for these families.

2-While I appreciate that the researchers cannot discriminate the size of the expansions, I wonder if they could assess any differences in "extremes of phenotype". For example, can they show differences in expansion in the samples from individuals who had the earliest onset versus those with the oldest onset. This would provide some preliminary evidence that the expansion was correlated with phenotype. Perhaps in keeping with this, were there any differences with average age of onset, by generation, to suggest anticipation.

3-The researchers assess STARD7 with RNA-Seq and show that there are no differences in expression. While I expect these to be "normal" I would have liked to have seen a few basic studies such as a western blot and/or localization experiments for STARD7. A downstream functional assessment of STARD7 would be ideal. I do agree that the normal expression, without the expansion, does suggest that the STARD7 is possibly not directly involved in disease pathogenesis and it is just a vehicle for this expansion to be expressed in the CNS.

4-The authors speculate that RNA toxicity is responsible for the disease given the evidence seen in another expansion-related disease, SCA37. The hypothesis was also based on the increased expression of genes involved in waste metabolism. The manuscript would benefit from tests to show a degree of RNA toxicity. Can the authors show that the cells have increased rate of cell death? Or perhaps RNA-FISH?

Reviewer #3:

Remarks to the Author:

This is a very strong paper and one that has answered a long-standing conundrum. The authors should be praised for this work.

Corbett et al. provide a good, succinct report building on previous studies of large families with epilepsy. This is relevant to neurogeneticists of epilepsy and epileptologists managing European families with FAME. I would recommend accepting this work.

This is a study of FAME2, a form of familial adult myoclonic epilepsy (FAME) linked to chr2 (most common linkage for European families with FAME). They used two forms of in silico analysis of WGS data from 6 individuals spanning 3 families to explore the FAME2 locus for an intronic combined insertion/expansion of the form found in FAME1, two other forms of FAME and SCA37. This they found in intron 1 of STARD7. They then screened members of these families plus another 19 families (most previously published) using repeat primed PCR and found that 158/160 affected individuals carried the genotype in question. The two non-carriers were felt to be phenocopies with a less severe clinical syndrome. The inserted expansion was absent in unrelated controls and databases. 2 clinically unaffected carriers in the families were young enough to not have manifested as yet, suggesting that overall within these families penetrance is high in late adulthood. Patient fibroblast studies suggested no change in STARD7 expression but enrichment analysis of differential gene expression suggested up-regulation of lysosomal and cell-death regulation pathways. They then interpret their findings in light of clinical and model work in other FAME genotypes and SCA37.

My main concern relates to the assertion that all 22 families have FAME2.

-page 8, line 172: I think the use of '22 independently reported FAME2 families' is misleading unless I have misunderstood table 1; not all 22 families have been independently reported, 16 have.

-I am concerned about the title of table 1 'Clinical summaries of 22 FAME families linked to chr2' as some of the families studied have not been demonstrated to have linkage to chr2 in the references supplied, and some have not previously been published so it is not known if other possible causes have been excluded.

Gardella, E. et al. Autosomal Dominant Early-onset Cortical Myoclonus, Photic-induced Myoclonus, and Epilepsy in a Large Pedigree. *Epilepsia* 47, 1643–1649 (2006) failed to find linkage to the FAME2 locus.

It is not clear if the family from Coppola, A. et al. Psychiatric comorbidities in patients from seven families with autosomal dominant cortical tremor, myoclonus, and epilepsy. *Epilepsy Behav.* EB 56, 38–43 (2016) is one of those previously demonstrated to have linkage to chromosome 2 (if so then the paper demonstrating this should be cited, rather than this purely phenotypic study).

No linkage study was performed in van Coller, R., van Rootselaar, A.-F., Schutte, C. & van der Meyden, C. H. Familial cortical myoclonic tremor and epilepsy: Description of a new South African pedigree with 30 year follow up. *Parkinsonism Relat. Disord.* 38, 35–40 (2017).

Abstract

How many affected individuals did not show segregation? (Answered later – put 158/160)

Are we going to use FAME as a familial description (a la GEFS+) or as a descriptor of an individual's epilepsy syndrome? So does the expansion segregate with FAME? Or does it segregate with typically affected individuals in FAME families?

Do you want to rephrase "These data, in combination with other genes bearing similar mutations that have been implicated in FAME, suggest ATTTTC expansions are the cause of this disorder, irrespective of the genomic locus involved." Which could mean - 'We predict that ATTTTC repeat expansions will be the cause of all future FAME families' – which is a pretty wild prediction.

Perhaps not inaccurate, but perhaps early to say.

Main text

The sentence "The frequency of GTCS varies from 15% to 100 % in different families and follows autosomal dominant inheritance" is a fragment as the subject of the sentence is 'frequency' here and frequency does not follow AD inheritance.

Use of both generalized and generalised in same manuscript

"rare individuals" or "rarely individuals"?

It is not clear how families are characterised as FAME1, FAME2 etc? If they have to have linkage first – how does this help the clinician? Are there are ethnicity or clinical features that separate the two?

What were the criteria for inclusion of families here? Presumably the six individuals were identified first and then a call was put out for further families – but how were these FAME2 pedigrees chosen? Did they all have linkage – how about 'family' 21 – the singleton?

I am concerned about the lack of expression of STARD7 in cases' cell-lines. I know that the authors have tried to square this circle, but this is with wishful thinking not with evidenced support. For example – what is the corollary with DAB1 cell-line expression in SCA?

Unanswered questions

Where does this leave the ADRA2B report from family 4, 15 etc?

Does the repeat expansion length predict the anticipation seen?

Which 'SC' has the conflict? Shreyasee Chakraborty or Shaun Carswell – presume the former from affiliations. Please clarify

Table 1

I am very grateful that the table is included, and it is important, but it really is a mish-mash where some families get ID reported, some get dementia reported and some have developmental delay reported. Ditto for psychiatric symptoms/disorders. Considering the large author list here of clinicians this should be ironed out.

Australian etc is not an ethnicity – unless you are telling me that the families are native Australians that predate Captain Cook. Suggest you mean nationality here but please comment on ethnicity.

Total affected is total number of individuals affected. Why are a small proportion of cases studied in some such as family 6?

Do the tremor and seizures start at a similar time? If not what is mean age of symptom onset telling us? Meaning these data will make them look more homogenous and falsely link the families.

What is a complex partial seizure with secondary generalization? Is this a focal seizure with impaired awareness leading to a bilateral tonic-clonic seizure?

Family 2 – 'Psychiatric symptoms' – as the prevalence of depressive and anxiety symptoms is so high in people with epilepsy – what is this telling us? Family 3 – OCD is a diagnosis not a symptom.

'Therapy resistant' 'Resistant convulsive seizures' ILAE has a definition of drug resistant epilepsy, can you tell me how many did and didn't meet this criterion? Use DRE as a unified term please.

Please can you look hard at the denominators here? For example - family 19, 13 studied and yet clinical features presented for 10.

Figure 1 – please can you separate panels a, b, c, d out so that they are easier to see? They are too bunched up.

Minor points:

-Figure 1 b) 'G' in 'AAATG' poorly visible

-page 10, line 211: incorrect position of references in text

Point by Point Responses to Reviews.

Editor's comments:

Please be aware that for certain types of new data, including most types of genetic data, journal policy is that deposition in a community-endorsed, public repository is generally mandatory prior to publication. Data submission can be a lengthy process, and we strongly suggest that you begin this well in advance of potential publication to avoid delays later on. Please include a statement about data availability in your point-by-point letter accompanying your revisions.

Please see decision letter above for more detailed information about data requirements and policy. If you are unable to make your data publically available for exceptional reasons, please get in touch with me now to discuss this further.

Author Response: *We have added a Data Availability statement.*

Reviewers' comments:

Reviewer #1 (Remarks to the Author):

In this manuscript by Gecz et al, they describe a new complex ATTTT/ATTTC pentanucleotide repeat expansion in intron 1 of STARD7 in a series of families with FAME. This is an important finding and builds on recent work demonstrating ATTTC repeat expansions at three other loci in FAME/BAFME families. Overall, this finding is important and the work is well done. The evidence supporting this repeat as the causative agent is compelling in light of the other recently published work with a few minor issues noted below. However, the manuscript itself requires some editing, with inclusion and expansion of some of the data already collected into the main body of the manuscript. In addition, while an expansive evaluation of how the repeat causes disease is beyond the scope of this manuscript, a few basic experiments should be done to better characterize how this repeat might cause disease.

Issues:

1) The two "false negative" cases need more characterization, as these post the greatest potential challenge to these ATTTC repeats being the causative element. A few questions related to these cases:

a. Do these cases carry the same disease associated haplotype region as the repeat that runs in their families or are they negative for the disease haplotype independent of their repeat findings?

Author Response: *We have now run chromosome 2 markers on these two individuals and they do not share the same haplotype as the affected individuals (new figure S3).*

b. Are either of these cases obligate causative gene carriers?

Author Response: *These are related individuals and are not obligate carriers.*

c. What is the evidence that they were phenocopies? Did they have the same EEG findings or just a vague history of myoclonus?

Author Response: *Diagnosis of FAME can be difficult as symptoms are subtle in some cases. These individuals (III-13 and IV-65) are a mother and daughter from the Australian / New Zealand family who did not have EEG examination or tonic-clonic seizures. We have always regarded their affected status as dubious. On the day of examination III-13 was unwell with an intercurrent illness. She was on quite a high dose of prednisolone and denied that she had a significant history of tremor. IV-65 had a self-reported history of tremor but evidence for this was not strong on examination. We published these individuals as affected in our original study¹, but excluded them from linkage analysis and from our later fine mapping work² because of their uncertain affected status. For completeness we included them in this expansion study however, in retrospect, it is not surprising that they are negative for the expansion.*

d. If they also had epilepsy, was there any other causative gene in these cases (and not in other cases) that might reveal the underlying cause of their diseases to allow us to more easily discount them?

Author Response: *They did not have epilepsy.*

2) The abstract suggests that RNA-seq analysis revealed involvement of protein quality control pathways (the preferred term for the autophagy and ubiquitin proteasome system, rather than "Cellular waste collection"). However, there is only a reference to three supplemental tables to justify this statement and these table are difficult to negotiate. If it is important enough to be in the abstract and discussed in the text, then it would be more appropriate for at least a summary of this RNA-seq data and GO analysis to be included within the main body of the manuscript- likely as part of a new figure.

Author Response: *We cultured four controls sourced from donors that do not have neurodevelopmental disorders and four fibroblasts from the Australian family during the recent review period and tested expression of ASAH1, PSAP, NPC2, TPP1 and GRN that we previously found differentially expressed, are expressed in the lysosome and are all involved in progressive myoclonic epilepsy. We could not replicate the gene expression differences from the RNA-seq data by qPCR in fibroblast cell lines cultured independently of the original cell lines used for RNA-seq. Were these cell responses due to the ATTC expansion, we would have expected to have seen similar expression changes in the genes we selected regardless of culture time and conditions however this was not the case. Given this uncertainty, we have removed the RNA-seq data from the manuscript and we thank the reviewer for questioning this.*

3) If you want to propose that there are alterations in protein quality control pathways, then this will require more robust analysis than what has been done so far. Specifically, the Gene ontology analysis need validation of the RNA-seq data by RT-PCR and (ideally) protein levels by immunoblot for key factors driving the GO analysis findings.

Author Response: *We were not able to validate these findings by qPCR and RNA-seq data have been removed. We shall be further investigating this in future work.*

4) Mature STARD7 mRNA levels should be independently confirmed by RT-PCR using primers that cross exon-exon junctions to determine the abundance of the mature mRNA.

Author Response: *We quantified levels of STARD7 RNA using primers spanning the repeat containing intron from exons 1 to 2 and a second pair spanning between exons 3 and 4. In both cases, there is no significant difference between the levels of STARD7 in individuals with the repeat when compared to control samples (see new figure 3a).*

5) The lack of repeat sequences within the RNA-seq dataset does not mean that the RNA is not accumulating as a) If they used oligodT primers for cDNA generation then they will miss any intronic sequences that were not included in the mature mRNA b) Repeat RNAs are often retained in the nucleus, so if their RNA isolation protocol included a spin that eliminated the nucleus prior to RNA isolation, it might be missed and c) the repeats may simply not be readable by RNA-seq given their size and repetitive highly AT-rich nature. There are no controls shown to demonstrate that they could even detect this RNA if it was present. Is the surrounding intronic mRNA sequence enriched/observed? Some evaluation of whether the repeat containing intron is retained (either within the RNA-seq data or by using intron/exon junction primer sets or with RNA foci determination by smFISH) in patient cells would be helpful and would provide some information about how the repeat might lead to disease.

Author Response: *RNA extractions were performed as described in the supplementary methods and do not include a centrifugation step prior to cell lysis. Our original extractions were mRNA libraries (TruSeq v2 unstranded library prep). In these samples, 5% of mapped reads were intronic. To specifically address the question of repeat containing reads, we have subsequently made total RNA extracts from fibroblasts and prepared them with the TruSeq Stranded total RNA library preparation kit with ribosomal RNA reduction. This protocol retains intronic reads. These two samples from Family 1, were sequenced to high depth with 529,688,759 and 444,894,773 mapped reads per sample. We detected 13% and 16% of mapped reads in introns in each sample respectively compared with 2% of reads in both samples mapped to intergenic regions. We do not find mapped or unmapped reads filled with the ATTC repeat sequence in these samples by analysis with TRhist.³ The repeat containing intron is not significantly retained compared with other introns in STARD7. We conclude fibroblast cell lines as is, are not suitable to address the pathogenic mechanism due the repeat and further experiments beyond the scope of our current report are required. The expression of the repeat may be different in the non-dividing cells of the brain as was observed for FAME1.⁴*

6) The STARD7 protein levels should be measured in patient cells compared to controls as a second measure for influence of the repeat on this gene.

Author Response: *Western blotting of whole protein extracts of fibroblasts from individuals with FAME showed no difference in protein expression compared to controls. Fibroblasts were cultured as described in Supplementary methods then lysed with lysis buffer (150mM NaCl, 1% Triton X-100, 1mM EDTA, 0.25% Sodium deoxycholate, 50mM Tris. Added protease inhibitor, NaF and Na₃VO₄). Extracts were separated by 4-12% polyacrylamide gel and transferred to nitrocellulose membrane by electroblotting. STARD7*

was detected with rabbit polyclonal anti-human/mouse/rat STARD7 (Proteintech cat# 15689-1-AP) at 1:500 dilution followed by anti-rabbit IgG conjugated to horseradish peroxidase (HRP) (Dako cat# P0448). Enhanced chemiluminescent detection (Bio-Rad cat# 1705061) was visualised with the chemidoc detection system (Bio-Rad). These data appear in the new figure 3b.

Minor points:

1) It is unclear why there is only one figure but a huge number of supplemental figures, many of which would be better in the main body of the manuscript. Figures S3, S4, and S5 contain information that will be of interest to the general reader, as would some visual representation of the data from tables S3-S6. This would make it easier for the readers to negotiate the manuscript and understand the findings and studies done.

Author Response: *The article was submitted as a brief communication however we welcome the chance to expand on the work. We made three new figures highlighting long read alignments and the gene expression.*

2) Is the repeat on a single haplotype across all of the families?

Author Response: *Based on our previous work, we believe it is likely that FAME2 maps to different haplotypes in different families other than those Italian families with previously documented founder effect.²*

3) Is there any evidence of anticipation in these families in terms of age of onset?

Author Response:

We have looked at this in the large Australian / New Zealand family and we do find evidence of anticipation. We have added the following to the manuscript: "In view of the discovery that FAME2 and FAME1 are caused by similar dynamic mutations of ATTC repeats, and the demonstration of clinical anticipation in FAME1,¹¹ we searched for evidence of anticipation in our pedigrees. We examined the median onset age of any relevant symptom, where available, for each generation in the Australian/New Zealand family (Family 1). We found evidence of anticipation; generation III had a median onset of 30 years (range 14-60y, n=6), in generation IV median onset was 17 years (8-50y, n=30) and the median onset in generation V was 12 years (4-19y, n=16). The remaining families were either too small or onset data was unavailable for anticipation to be robustly assessed."

Reviewer #2 (Remarks to the Author):

Corbett et al identify an intronic expansion mutation in STARD7 as the cause of familial adult myoclonic epilepsy type 2 (FAME2; linked to chromosome 2).

The researchers analysed WGS data from 6 individuals from 3 families. They searched for TTTTA and TTTC expansions within the known linkage region of chromosome 2 using ExpansionHunter and exSTRa. They identified an expansion of the endogenous ATTTT as well as an insertion of an expanded ATTTT within the STARD7 intron. They then performed an assay to screen 160 individuals from 22 families linked to FAME2 locus.

They showed the presence of the insertion expansion (ATTTC) in 158/160 and it was not present in controls. RNA studies showed that STARD7 was expressed similarly among 6 cases and controls. RNA-Seq showed differential expression of several hundred genes across the genome. Genes involved in the metabolism of waste products appeared to be enriched.

The manuscript presents highly convincing evidence that the STARD7 expansion is responsible for FAME2. This is a robust and novel finding. The quality of the writing is excellent and the methods are sound. Results are interpreted appropriately. The authors are leaders in the field of epilepsy genetics.

Comments:

1) Family 4 and 15 were previously published as having FAME2 due to mutation in ADRA2B. While I see that they comment on this variant in their Table, can the authors comment, in the text, that this molecular diagnosis was incorrect for these families.

Author Response: *The ADRA2B variant was reported by de Fusco et al. ⁵ as a FAME associated gene. It was shown that the variant had functional effects on Ca²⁺ signalling. We have added a new sentence to the text acknowledging this allele is not causative.*

2) While I appreciate that the researchers cannot discriminate the size of the expansions, I wonder if they could assess any differences in “extremes of phenotype”. For example, can they show differences in expansion in the samples from individuals who had the earliest onset versus those with the oldest onset? This would provide some preliminary evidence that the expansion was correlated with phenotype. Perhaps in keeping with this, were there any differences with average age of onset, by generation, to suggest anticipation?

Author Response: *Please refer to comments above re: anticipation in the response to Reviewer #1, minor point #3. We now have a second read from our reanalysed Oxford Nanopore data showing heterogeneity of repeat size within a DNA sample extracted from a single patient cell line. We have modified the main text to describe these results include a new figure 2 and have also added these sequences to the Supplementary Data.*

3) The researchers assess STARD7 with RNA-Seq and show that there are no differences in expression. While I expect these to be “normal” I would have liked to have seen a few basic studies such as a western blot and/or localization experiments for STARD7. A downstream functional assessment of STARD7 would be ideal. I do agree that the normal expression, without the expansion, does suggest that the STARD7 is possibly not directly involved in disease pathogenesis and it is just a vehicle for this expansion to be expressed in the CNS.

Author Response: *We have confirmed STARD7 expression is not altered by western blot and by relative qPCR with multiple primer pairs to STARD7 (new Figure 3). Please refer also to our response to Reviewer #1 major question #4.*

4) The authors speculate that RNA toxicity is responsible for the disease given the evidence seen in another expansion-related disease, SCA37. The hypothesis was also based on the increased expression of genes involved in waste metabolism. The

manuscript would benefit from tests to show a degree of RNA toxicity. Can the authors show that the cells have increased rate of cell death? Or perhaps RNA-FISH?

Author Response: *We were not able to replicate these findings from RNA-seq in cell lines we cultured to address these concerns and those of Reviewer #1. Given the uncertainty of using these fibroblasts to make conclusions about disease mechanism we propose that a separate and specific investigation into mechanism is required. This work is beyond the scope of the strong genetic data that we present and requires a carefully designed model system (e.g. iPS derived neurons) and / or tissue biopsy from patient brain to address.*

Reviewer #3 (Remarks to the Author):

This is a very strong paper and one that has answered a long-standing conundrum. The authors should be praised for this work.

Corbett et al. provide a good, succinct report building on previous studies of large families with epilepsy. This is relevant to neurogeneticists of epilepsy and epileptologists managing European families with FAME. I would recommend accepting this work.

This is a study of FAME2, a form of familial adult myoclonic epilepsy (FAME) linked to chr2 (most common linkage for European families with FAME). They used two forms of in silico analysis of WGS data from 6 individuals spanning 3 families to explore the FAME2 locus for an intronic combined insertion/expansion of the form found in FAME1, two other forms of FAME and SCA37. This they found in intron 1 of STARD7. They then screened members of these families plus another 19 families (most previously published) using repeat primed PCR and found that 158/160 affected individuals carried the genotype in question. The two non-carriers were felt to be phenocopies with a less secure clinical syndrome. The inserted expansion was absent in unrelated controls and databases. 2 clinically unaffected carriers in the families were young enough to not have manifest as yet, suggesting that overall within these families penetrance is high in late adulthood. Patient fibroblast studies suggested no change in STARD7 expression but enrichment analysis of differential gene expression suggested up-regulation of lysosomal and cell-death regulation pathways. They then interpret their findings in light of clinical and model work in other FAME genotypes and SCA37.

My main concern relates to the assertion that all 22 families have FAME2.

-page 8, line 172: I think the use of '22 independently reported FAME2 families' is misleading unless I have misunderstood table 1; not all 22 families have been independently reported, 16 have.

-I am concerned about the title of table 1 'Clinical summaries of 22 FAME families linked to chr2' as some of the families studied have not been demonstrated to have linkage to chr2 in the references supplied, and some have not previously been published so it is not known if other possible causes have been excluded.

Author Response: *We have amended the title of this table and also added levels of evidence for linkage to chromosome 2 for all families.*

Gardella, E. et al. Autosomal Dominant Early-onset Cortical Myoclonus, Photic-induced

Myoclonus, and Epilepsy in a Large Pedigree. *Epilepsia* 47, 1643–1649 (2006) failed to find linkage to the FAME2 locus.

Author Response: *The original linkage analysis⁶ was performed on several individuals that were examined at younger than ten years of age and typed as affected based on neurological examinations indicating rhythmic cortical myoclonus but myoclonic jerks were not evident. In figure S2f we show segregation of markers mapping to the known chromosome 2 interval, furthermore affected individuals share the chromosome 2 haplotype with known linked Southern Italian families (this is Family 3 in Henden et al. but it was not referenced).²We have now added this information to Table 1.*

It is not clear if the family from Coppola, A. et al. Psychiatric comorbidities in patients from seven families with autosomal dominant cortical tremor, myoclonus, and epilepsy. *Epilepsy Behav.* EB 56, 38–43 (2016) is one of those previously demonstrated to have linkage to chromosome 2 (if so then the paper demonstrating this should be cited, rather than this purely phenotypic study).

Author Response: *We have quoted the original work mapping the FAME2 locus in the families described in Coppola et al., 2016.⁷ The 2016's paper by Coppola et al did also include a previously unreported family linked to the FAME 2 locus. Therefore, it is properly quoted into the table.*

No linkage study was performed in van Coller, R., van Rootselaar, A.-F., Schutte, C. & van der Meyden, C. H. Familial cortical myoclonic tremor and epilepsy: Description of a new South African pedigree with 30 year follow up. *Parkinsonism Relat. Disord.* 38, 35–40 (2017).

Author Response: *The lack of linkage data for this family is now indicated in the table. The family clinically fits the picture of FAME.*

Abstract

How many affected individuals did not show segregation? (Answered later – put 158/160).

Author Response: *This has been amended to 137/137 individuals from families with some evidence of chr2 linkage. Based on the helpful comments of this reviewer we have substantially revised the presentation of the numbers of individuals we screened as discussed below.*

Are we going to use FAME as a familial description (a la GEFS+) or as a descriptor of an individual's epilepsy syndrome? So does the expansion segregate with FAME? Or does it segregate with typically affected individuals in FAME families?

Author Response: *The term FAME should be used as a familial description rather than a descriptor of an individual's epilepsy syndrome. Accordingly, we amended this line to say the expansion segregates with typically affected individuals in FAME families.*

Do you want to rephrase "These data, in combination with other genes bearing similar mutations that have been implicated in FAME, suggest ATTTC expansions are the cause of this disorder, irrespective of the genomic locus involved?" Which could mean - 'We predict that ATTTC repeat expansions will be the cause of all future FAME families' –

which is a pretty wild prediction. Perhaps not inaccurate, but perhaps early to say.

Author Response: *Given that the three major loci chr2, 5 & 8 which account for the majority of all reported families have this expansion and the two new genes identified by Ishihura et al. also have this identical expansion, we think the evidence that expansion of ATTTC repeat is a general cause of the disease is strong. This is important as it can facilitate discovery of as yet unknown loci. We amended the text slightly to: "suggest ATTTC expansions **may** cause this disorder, irrespective of the genomic locus involved." This prediction therefore does not preclude alternative repeat sequences having a role in FAME.*

Main text

The sentence "The frequency of GTCS varies from 15% to 100 % in different families and follows autosomal dominant inheritance" is a fragment as the subject of the sentence is 'frequency' here and frequency does not follow AD inheritance.

Author Response: *Amended to: The frequency of GTCS varies from 15% to 100 % in different families (Table 1).*

Use of both generalized and generalised in same manuscript

"rare individuals" or "rarely individuals"?

Author Response: *We have checked the manuscript for grammatical and spelling errors and made corrections where appropriate.*

It is not clear how families are characterised as FAME1, FAME2 etc.? If they have to have linkage first – how does this help the clinician? Are there are ethnicity or clinical features that separate the two?

Author Response: *This is an insightful comment. To the best of our knowledge, there are no obvious clinical features to separate FAME1, 2 or 3. To date it appears chr8 is predominantly Asian while chr2 and chr5 are predominantly European although chr5 linkage has been reported in one Asian family.⁸ In this current manuscript, we showed the presence of an ATTTC repeat expansion in STARD7 in families of Israeli (Sephardic Jewish) and Arab descent (from Syria). For a molecular genetic test, cascading Asian families for chr8 (SAMD12) and European families for chr2 then chr5 before testing all other known loci would be the best strategy. We added the following to the discussion: "The FAME2 locus is the most frequently observed linked region for Caucasian individuals affected by this disorder whereas chromosome 8 thus far is limited to Asian individuals, therefore molecular genetic testing should take this into consideration if choosing to screen by RP-PCR.*

What were the criteria for inclusion of families here? Presumably the six individuals were identified first and then a call was put out for further families – but how were these FAME2 pedigrees chosen? Did they all have linkage – how about 'family' 21 – the singleton?

Author Response: *The families were recruited through the FAME consortium with the larger families having definitive or suggestive linkage to the 2q locus. The initial detection of the repeat was in individuals for whom we had WGS data, as the reviewer has intuited.*

Additional individuals and small families were screened on the basis of a compatible clinical phenotype. We have substantially revised our description of our screening to help clarify this.

"We developed a repeat primed PCR (RP-PCR) assay to rapidly identify the expansion in 137/137 affected individuals from 16 independently reported FAME2 families worldwide (Fig. 1c and 1d, Table 1, Table S2, Fig. S2; Families 1, 3-6, 8-10, 12-16, 19, 20 & 22). Of the 24 individuals tested in these families that did not have a FAME diagnosis, 2 were positive for the ATTC expansion; both were from younger generations and likely presymptomatic. We tested an additional 72 individuals (52 unrelated and 20 cases from 6 families with multiple affected individuals) with clinical similarity to FAME. Of these, 20/20 familial and 1/52 singleton cases were positive for an ATTC expansion in STARD7 (Table 1, Fig. S2; Families 2, 7, 11, 17, 18 & 21 [singleton case]). The 52 unrelated subjects comprised 13 subjects with generalised epilepsy and tremor and 39 with myoclonic epilepsy with onset over the age of 19 years; 8/52 cases had a family history of epilepsy. Of 13 individuals that had uncertain diagnoses in all of the families we tested, 8 carried the ATTC expansion. Two of the individuals with uncertain diagnosis that tested negative were a mother and daughter pair from Family 1 (Fig. S2a [red box] III-13 and IV-65) and subsequent analyses with microsatellite markers showed that these individuals did not have the same haplotype as affected carriers of the ATTC expansion (Fig. S3). The ATTC repeat expansion did not amplify in any of 28 control DNA samples extracted from unaffected individuals unrelated to FAME."

I am concerned about the lack of expression of STARD7 in cases' cell-lines. I know that the authors have tried to square this circle, but this is with wishful thinking not with evidenced support. For example – what is the corollary with DAB1 cell-line expression in SCA?

Author Response: *STARD7 is expressed in both fibroblast and lymphoblastoid patient cell lines however we have not seen the repeat sequence expressed with the assays we have used to date. DAB1 is not expressed in fibroblasts (see new figure 4 that was formerly in supplementary data) and also figure 6E in Seixas et al.⁹ To look at cellular mechanism Seixas et al. over expressed a repeat of 58 consecutive ATTC repeats in a non-neuronal laboratory cell line and saw RNA foci by in situ hybridization.⁹ The presence of these RNA foci was informative given that the endogenous repeat did not cause that response however this is not necessarily the mechanism of disease pathogenesis. We are intrigued by the mechanism and further work is planned to address this.*

Unanswered questions

Where does this leave the ADRA2B report from family 4, 15 etc.?

Author Response: *This was also questioned by Reviewer #2. The ADRA2B variant is very likely benign with respect to FAME and this has now been specifically mentioned in the text.*

Does the repeat expansion length predict the anticipation seen?

Author Response: *We do not have sufficient molecular data to make this correlation. We have added new long read sequencing data and show that there is potential somatic*

instability of the repeat.

Which 'SC' has the conflict? Shreyasee Chakraborty or Shaun Carswell – presume the former from affiliations. Please clarify

Author Response: *We have amended this comment to explicitly name the individuals. The reviewer is correct.*

Table 1

I am very grateful that the table is included, and it is important, but it really is a mish-mash where some families get ID reported, some get dementia reported and some have developmental delay reported. Ditto for psychiatric symptoms/disorders. Considering the large author list here of clinicians this should be ironed out.

Author Response: *Terms and clinical data in the table have been harmonised where possible. In some cases, clinical data was not initially collected and it was not possible to reassess all in the time available during the review period.*

Australian etc. is not an ethnicity – unless you are telling me that the families are native Australians that predate Captain Cook. Suggest you mean nationality here but please comment on ethnicity.

Author Response: *The Australian / NZ family is of European descent. We have now amended Table 1 to make ancestry clear when it is not matched with the nationality.*

Total affected is total number of individuals affected. Why are a small proportion of cases studied in some such as family 6?

Author Response: *We tested all available DNA, many of these families have been ascertained over 20 years ago and have had many genetic investigations before we found the STARD7 mutation. It was not possible to recontact some of them to obtain new samples for DNA extraction.*

Do the tremor and seizures start at a similar time? If not what is mean age of symptom onset telling us? Meaning these data will make them look more homogenous and falsely link the families.

Author Response: *It is difficult to dissect out onset of the different FAME symptoms, especially retrospectively in the older generations. This is especially so for myoclonic seizures and tremor – indeed the tremor is neurophysiologically a form of cortical myoclonus. Further, not all cases have tonic-clonic seizures. Thus pragmatically, we have used age of onset of any relevant symptom (myoclonic seizures, tonic-clonic seizures or tremor) to mark the onset of disease.*

What is a complex partial seizure with secondary generalization? Is this a focal seizure with impaired awareness leading to a bilateral tonic-clonic seizure?

Author Response: *Correct, thank you. According to the new ILAE classification, this means a focal seizure with impaired awareness leading to a bilateral tonic-clonic seizure.*

Family 2 – ‘Psychiatric symptoms’ – as the prevalence of depressive and anxiety symptoms is so high in people with epilepsy – what is this telling us?

Author Response: *Data on psychiatric symptoms are provided for information only. Approximately one third of people with epilepsy have a psychiatric comorbidity, it does not appear from the data we have obtained that this is significantly different for FAME.*

Family 3 – OCD is a diagnosis not a symptom.

‘Therapy resistant’ ‘Resistant convulsive seizures’ ILAE has a definition of drug resistant epilepsy, can you tell me how many did and didn’t meet this criterion? Use DRE as a unified term please.

Author Response: *We specified that OCD includes reoccurring thoughts and compulsive behaviour. The term ‘Resistant convulsive seizures’ and ‘Therapy resistant seizures’ has been replaced with DRE (i.e., seizures not responsive to at least 2 trials with appropriate drugs).*

Please can you look hard at the denominators here? For example - family 19, 13 studied and yet clinical features presented for 10.

Author Response: *The numbers are correct for the categories where the dominators were changed, these measures were not made on all family members.*

Figure 1 – please can you separate panels a, b, c, d out so that they are easier to see? They are too bunched up.

Author Response: *We have added extra space to this figure and increased the sizes of the fonts.*

Minor points:

-Figure 1 b) ‘G’ in ‘AAATG’ poorly visible

Author Response: *This has been altered.*

-page 10, line 211: incorrect position of references in text

Author Response: *This has been corrected.*

References:

1. Crompton, D. E. *et al.* Familial adult myoclonic epilepsy: recognition of mild phenotypes and refinement of the 2q locus. *Arch. Neurol.* **69**, 474–481 (2012).
2. Henden, L. *et al.* Identity by descent fine mapping of familial adult myoclonus epilepsy (FAME) to 2p11.2-2q11.2. *Hum. Genet.* (2016). doi:10.1007/s00439-016-1700-8
3. Doi, K. *et al.* Rapid detection of expanded short tandem repeats in personal genomics using hybrid sequencing. *Bioinformatics* **30**, 815–822 (2014).

4. Ishiura, H. *et al.* Expansions of intronic TTTCA and TTTTA repeats in benign adult familial myoclonic epilepsy. *Nat. Genet.* **50**, 581–590 (2018).
5. De Fusco, M. *et al.* The $\alpha 2B$ -adrenergic receptor is mutant in cortical myoclonus and epilepsy. *Ann. Neurol.* (2013). doi:10.1002/ana.24028
6. Gardella, E. *et al.* Autosomal Dominant Early-onset Cortical Myoclonus, Photic-induced Myoclonus, and Epilepsy in a Large Pedigree. *Epilepsia* **47**, 1643–1649 (2006).
7. Coppola, A. *et al.* Psychiatric comorbidities in patients from seven families with autosomal dominant cortical tremor, myoclonus, and epilepsy. *Epilepsy Behav* **56**, 38–43 (2016).
8. Li, J. *et al.* A Chinese benign adult familial myoclonic epilepsy pedigree suggesting linkage to chromosome 5p15.31-p15.1. *Cell Biochem. Biophys.* **69**, 627–631 (2014).
9. Seixas, A. I. *et al.* A Pentanucleotide ATTTTC Repeat Insertion in the Non-coding Region of DAB1, Mapping to SCA37, Causes Spinocerebellar Ataxia. *Am J Hum Genet* **101**, 87–103 (2017).

Reviewers' Comments:

Reviewer #1:

None

Reviewer #2:

Remarks to the Author:

I have no further remarks for the authors. I feel they have addressed my concerns, as well as the other reviewers, and that this is a strong paper that provides a genetic explanation for FAME2. I am happy to see they have performed a few further experiments but I agree with the authors that an elucidation of the molecular mechanism is beyond the scope of this manuscript.

Reviewer #3:

Remarks to the Author:

I am very impressed by the way the authors have clearly and comprehensively answered the range of queries raised from the three reviewers. These comments have significantly strengthened the paper and I have no more to add. They should be congratulated for their diligence here.

REVIEWERS' COMMENTS:

Reviewer #1 made no further comments

Reviewer #2 (Remarks to the Author):

I have no further remarks for the authors. I feel they have addressed my concerns, as well as the other reviewers, and that this is a strong paper that provides a genetic explanation for FAME2. I am happy to see they have performed a few further experiments but I agree with the authors that an elucidation of the molecular mechanism is beyond the scope of this manuscript.

Reviewer #3 (Remarks to the Author):

I am very impressed by the way the authors have clearly and comprehensively answered the range of queries raised from the three reviewers. These comments have significantly strengthened the paper and I have no more to add. They should be congratulated for their diligence here.